# Learning under uncertainty: a comparison between R-W and Bayesian approach

**He Huang**
Laureate Institute for Brain Research
Tulsa, OK, 74133
crane081@gmail.com

**Martin Paulus**
Laureate Institute for Brain Research
Tulsa, OK, 74133
mpaulus@laureateinstitute.org

## Abstract

Accurately differentiating between what are truly unpredictably random and systematic changes that occur at random can have profound effect on affect and cognition. To examine the underlying computational principles that guide different learning behavior in an uncertain environment, we compared an R-W model and a Bayesian approach in a visual search task with different volatility levels. Both R-W model and the Bayesian approach reflected an individual's estimation of the environmental volatility, and there is a strong correlation between the learning rate in R-W model and the belief of stationarity in the Bayesian approach in different volatility conditions. In a low volatility condition, R-W model indicates that learning rate positively correlates with lose-shift rate, but not choice optimality (inverted U shape). The Bayesian approach indicates that the belief of environmental stationarity positively correlates with choice optimality, but not lose-shift rate (inverted U shape). In addition, we showed that comparing to Expert learners, individuals with high lose-shift rate (sub-optimal learners) had significantly higher learning rate estimated from R-W model and lower belief of stationarity from the Bayesian model.

## 1   Introduction

Learning and using environmental statistics in choice-selection under uncertainty is a fundamental survival skill. It has been shown that, in tasks with embedded environmental statistics, subjects use sub-optimal heuristic Win-Stay-Lose-Shift (WSLS) strategy (Lee et al. 2011), and strategies that can be interpreted using Reinforcement Learning model (Behrens et al. 2007), or Bayesian inference model (Mathys et al. 2014; Yu et al. 2014). Value-based model-free RL model assumes subjects learn the values of chosen options using a prediction error that is scaled by a *learning rate* (Rescorla and Wagner, 1972; Sutton and Barto, 1998). This learning rate can be used to measure an individual's reaction to environmental volatility (Browning et al. 2015). Higher learning rate is usually associated with a more volatile environment, and a lower learning rate is associated with a relatively stable situation. Different from traditional (model-free) RL model, Bayesian approach assumes subjects make decisions by learning the reward probability distribution of all options based on Bayes' rule, i.e., sequentially updating the posterior probability by combining the prior knowledge and the new observation (likelihood function) over time. To examine how environment volatility may influence this inference process, Yu & Cohen 2009 proposed to use a dynamic belief model (DBM)

that assumes subjects update their belief of the environmental statistics by balancing between the prior belief and *the belief of environmental stationarity*, in which a belief of high stationarity will lead to a relatively fixed belief of the environmental statistics, and vice versa.

Though formulated under different assumptions (Gershman 2015), those two approaches share similar characteristics. First, both the learning rate in RL model and the belief of stationarity in DBM reflect an individual's estimation of the environmental volatility. In a highly volatile environment, one will be expected to have a high learning rate estimated by RL model, and a belief of low stationarity estimated by DBM. Second, though standard RL only updates the chosen option's value and DBM updates the posterior probability of all options, assuming maximization decision rule (Blakely, Starin, & Poling, 1988), both models will lead to qualitatively similar choice preference. That is, the mostly rewarded choice will have an increasing value in RL model, and increasing reward probability in DBM, while the often-unrewarded choice will lead to a decreasing value in RL model, and decreasing reward probability in DBM. Thirdly, they can both explain Win-Stay strategy. That is, under the maximization assumption, choosing the option with maximum value in RL model, the rewarded option (Win) will reinforce this choice (i.e. remain the option with the maximum value) and thus will be chosen again (Stay). Similarly, choosing the option with the maximum reward probability in DBM, the rewarded option (Win) will also reinforce this choice (i.e. remain the option with the maximum reward probability) and thus will be chosen again (Stay).

While both approaches share some characteristics as mentioned above and have showed strong evidence in explaining the overall subjects' choices in previous studies, it is unclear how they differ in explaining other behavioral measures in tasks with changing reward contingency, such as *decision optimality*, i.e., percentage of trials in which one chooses the most likely rewarded option, and *lose-shift rate*, i.e., the tendency to follow the last target if the current choice is not rewarded. In a task with changing reward contingency (e.g., 80%:20% to 20%:80%), decision optimality relies on proper estimation of the environmental volatility, i.e., how frequent change points occur, and using proper strategy, i.e. staying with the mostly likely option and ignoring the noise before change points (i.e. not switching to the option with lower reward rate when it appears as the target). Thus it is important to know how the parameter in each model (learning rate vs. the belief of stationarity) affects decision optimality in tasks with different volatility. On the other hand, lose-shift can be explained as a heuristic decision policy that is used to reduce a cognitively difficult problem (Kahneman & Frederick, 2002), or as an artifact of learning that can be interpreted in a principled fashion (using RL: Worthy et al. 2014; using Bayesian inference: Bonawitz et al. 2014). Intuitively, when experiencing a loss in the current trial, in a high volatility environment where change points frequently occur, one may tend to shift to the last target; while in a stable environment with fixed reward rates, one may tend to stay with the option with the higher reward rate. That is, the frequency of using lose-shift strategy should depend on how frequent the environment changes. Thus it is also important to examine how the parameter in each model (learning rate vs. the belief of stationarity) affects lose-shift rate under different volatility conditions.

However, so far little is known about how a model-free RL model and a Bayesian model differ in explaining decision optimality and lose-shift in tasks with different levels of volatility. In addition, it is unclear if parameters in each model can capture the individual differences in learning. For example, if they can provide satisfactory explanation of individuals who always choose the same choice while disregarding feedback information (No Learning), individuals who always choose the most likely rewarded option (expert), and individuals who always use the heuristic win-stay-lose-shift strategy. Here we aim to address the first question by investigating the relationship between decision optimality and lose-shift rate with parameters estimated from an Rescorla-Wagner (R-W model) and a Bayesian model in three volatility conditions (Fig 1a) in a visual search task (Yu et al. 2014): 1) **stable**, where the reward contingency at three locations remains the same (relative reward frequency at three locations: 1:3:9), 2) **low volatility**, where the reward contingency at three locations changes (e.g. from 1:3:9 to 9:1:3) based on $N(30, 1)$ (i.e. on average change points occur every 30 trials), and 3) **high volatility**, where the reward contingency changes based on $N(10, 1)$ (i.e. on average change points occur every 10 trials). For the second question, we will examine how the two models differ in explaining three types of behavior: *No Learning*, *Expert*, and *WSLS* (Fig 1b).

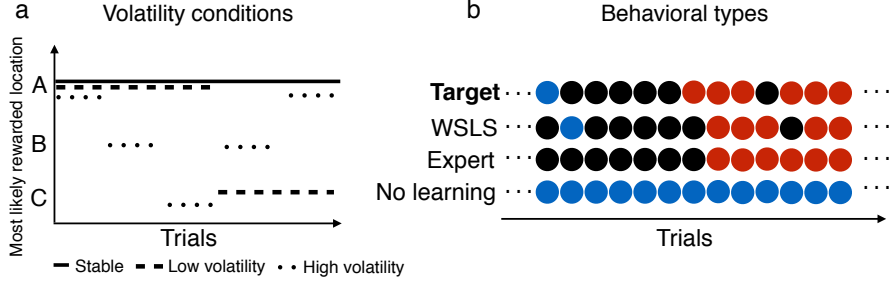

Figure 1: Example of volatility conditions and behavioral types in a visual search task. a. Example of three volatility conditions. b. Example of three behavioral types. Colors indicate target location in each trial. *WSLS*: Win-stay-Lose-shift (follow last target). *Expert*: always choose the most likely location. *No Learning*: always choose the same location.

## 2  Value-based RL model

Assuming a constant learning rate, the Rescorla-Wagner model takes the following form (Rescorla and Wagner, 1972; Sutton and Barto, 1998):

$$V_i^{t+1} = V_i^t + \eta(R_t - V_i^t) \tag{1}$$

where $\eta$ is the learning rate, and $R_t$ is the reward feedback (0-no reward, 1-reward) for the chosen option $i$ in trial $t$. In this paper, we assume subjects use a softmax decision rule for all models as follows:

$$p(i) = \frac{e^{\beta V_i}}{\sum_j e^{\beta V_j}} \tag{2}$$

where $\beta$ is the inverse decision temperature parameter, which measures the degree to which subjects use the estimated value in choosing among three options (i.e. a large $\beta$ approximates 'maximization' strategy). This model has two free parameters = $\{\eta, \beta\}$ for each subject.

### 2.1  Simulation in three volatility conditions

Learning rate is expected to increase as the volatility increases. To show this, we simulated three volatility conditions (*Stable*, *Low* and *High volatility*, Fig 2ab) and the results are summarized in Table1. For each condition, we simulated 100 runs (90 trial per run) of agents' choices with $\eta$ ranges from 0 to 1 with an increment of 0.1 and fixed $\beta = 20$. As is shown in Fig 2a, decision optimality in a stable and a low volatility environment has an inverted U shape as a function of learning rate $\eta$. It is not surprising, as in those conditions, where one should rely more on the long term statistics, if the learning rate is too high, then subjects will tend to shift more due to recent experience (Fig 2b), which would adversely influence decision optimality. On the other hand, in a high volatility environment, decision optimality has a linear correlation with the learning rate, suggesting that higher learning rate leads to better performance. In fact, the optimal learning rate increases as the environmental volatility increases (i.e. the peak of the inverted U should shift to the right). On the other hand, across all volatility conditions, lose shift rate increases as learning rate increases (Fig 2b), except for learning rate=0. It is not surprising as zero learning rate indicates subjects make random choices, thus it will be close to $1/3$.

### 2.2  Simulation of three behavioral types

To examine if learning rate can be used to explain different types of learning behavior, we have simulated three types of behavior (*No Learning, Expert and WSLS*, Fig 1b) in a low volatility condition. In particular, we simulated 60 runs (90 trials per run) of target sequences with a relative reward frequency 1:3:9 that changes based on $N(30, 1)$, and generated three types of behavior for

Table 1: R-W model: Influence of learning rate $\eta$

| Condition | Decision optimality | Lose-shift rate |
|---|---|---|
| Stable | Inverted U shape, $\eta_{optimal}$ = low | Positive linear |
| Low volatility | Inverted U shape, $\eta_{optimal}$ = medium | Positive linear |
| High volatility | Positive linear relationship, $\eta_{optimal}$ = high | Positive linear |

each run. For each simulated behavior type, R-W model was fitted using Maximum Likelihood Estimation with $\eta$ ranges from 0 to 1 with an increment of .025 and $\beta = 20$. Based on what we have shown in 2.1, in a low volatility condition where decision optimality has an inverted U shape as a function of learning rate, individuals that perform poorly will be expected to have a low learning rate, and individuals that use heuristic WSLS strategy will be expected to have a high learning rate. We confirmed this in simulation (Fig 2c), that agents with the same choice over time (*No Learning*) have the lowest learning rate, indicating their choices have little influence from the reward feedback. *Expert* agents have the medium learning rate indicating the effect of long-term statistics. Agents that strictly follow *WSLS* have the highest learning rate, indicating their choices are heavily impacted by recent experience. Results for learning rate estimation of three behavioral types in stable and high volatility condition can be seen in Supplementary Figure S1.

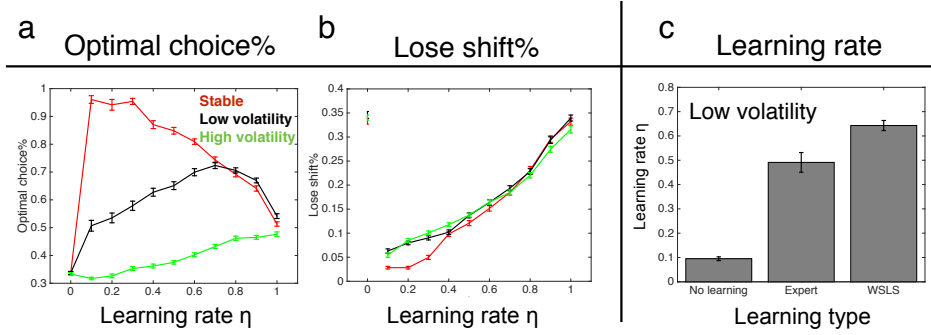

Figure 2: R-W simulation. a. Percentage of trials in which agents chose the optimal choice (the most likely location) as a function of learning rate in three volatility conditions. b. Lose shift rate as a function of learning rate in three volatility conditions. c. Learning rate estimation of three simulated behavior types in low volatility condition. Errorbars indicate standard error of the mean across simulation runs.

## 3   A Bayesian approach

Here we compare above R-W model to a dynamic belief model that is based on a Bayesian hidden Markov model, where we assume subjects make decisions by using the inferred posterior target probability $\mathbf{s}_t$ based on the inferred hidden reward probability $\boldsymbol{\gamma}_t$ and the reward contingency mapping $b_t$ (Equation 3). To examine the influence of volatility, we assume $(\boldsymbol{\gamma}_t, b_t)$ has probability $\alpha$ of remaining the same as the last trial, and $1-\alpha$ of being drawn from the prior distribution $p_0(\boldsymbol{\gamma}_t, b_t)$ (Equation 4). Here $\alpha$ represents an individual's estimation of environmental stationarity, which contrasts with learning rate $\eta$ in R-W model. For model details, please refer to Yu & Huang 2014.

$$P(\mathbf{s}_t|\boldsymbol{\gamma}_t, b_t) = \begin{cases} (\frac{1}{3}, \frac{1}{3}, \frac{1}{3}), & b_k = 1 \\ (\gamma_h, \gamma_m, \gamma_l), & b_k = 2 \\ (\gamma_h, \gamma_l, \gamma_m), & b_k = 3 \\ (\gamma_m, \gamma_h, \gamma_l), & b_k = 4 \\ (\gamma_m, \gamma_l, \gamma_h), & b_k = 5 \\ (\gamma_l, \gamma_h, \gamma_m), & b_k = 6 \\ (\gamma_l, \gamma_m, \gamma_h), & b_k = 7 \end{cases} \tag{3}$$

$$P(\boldsymbol{\gamma}_t, b_t | \mathbf{s}_{t-1}) = \alpha P(\boldsymbol{\gamma}_{t-1}, b_{t-1} | \mathbf{s}_{t-1}) + (1 - \alpha) p_0(\boldsymbol{\gamma}_t, b_t) \tag{4}$$

where $\boldsymbol{\gamma}_t$ is the hidden reward probability, $b_t$ is the reward contingency mapping of the probability to the options, $\mathbf{s}_{t-1}$ is the target history from trial 1 to trial $t - 1$. We also used softmax decision rule here (Equation 2), thus this model also has two free parameters = $\{\alpha, \beta\}$.

## 3.1 Simulation in three volatility conditions

Belief of stationarity $\alpha$ is expected to decrease as the volatility increases, as subjects are expected to depend more on the recent trials to predict next outcome (Yu &Huang 2014). We have shown this in three simulated conditions under different volatility (Fig 3ab) and the results are summarized in Table 2. For each simulated condition (stable, low and high volatility), we simulated 100 runs (90 trials per run) for agents' choices with $\alpha$ ranges from 0 to 1 and fixed $\beta = 20$. As is shown in Fig 3a, in a stable condition, decision optimality increases as $\alpha$ increases, indicating a fixed belief mode (i.e. no change of the environmental statistics) is optimal in this condition. In the other two volatile environments, decision optimality also increases as $\alpha$ increases, but both drop as $\alpha$ approaches 1. It is reasonable as in volatile environments a belief of high stationarity is no longer optimal. On the other hand, lose shift rate in all conditions (Fig 3b) have an inverted U shape as a function of alpha, where $\alpha = 0$ leads to a random lose shift rate ($1/3$), and $\alpha = 1$(fixed belief model) leads to the minimal lose shift rate.

## 3.2 Simulation of three behavioral types

To examine if an individual's belief of environmental stationarity can be used to explain different types of learning behavior, we fit DBM using Maximum Likelihood Estimation with the simulated behavioral data in 2.2. DBM results suggest that *WSLS* has a significantly lower belief of stationarity comparing to *Expert* behavior (Fig 3c), which is consistent with the higher volatility estimation reflected by a higher learning rate than *Expert* from R-W model (Fig 2c). Simulation results also suggest that *No Learning* agents have a significantly lower belief of stationarity than *Expert* learners, but not different from *WSLS*. However, the comparison of model accuracy between R-W and DBM (Fig 3d) shows DBM outperforms R-W in predicting *Expert* and *WSLS* behavior ($p = .000$), but it does not perform as well in *No Learning* behavior where R-W has significantly better performance ($p = .000$). Model accuracy is measured as the percentage of trials that the model correctly predicted subjects' choice. Thus further investigation is needed to examine the validity of using DBM in explaining poor learners' choice behavior in this task. Results for $\alpha$ in stable and high volatility condition can be seen in Supplementary Figure S2.

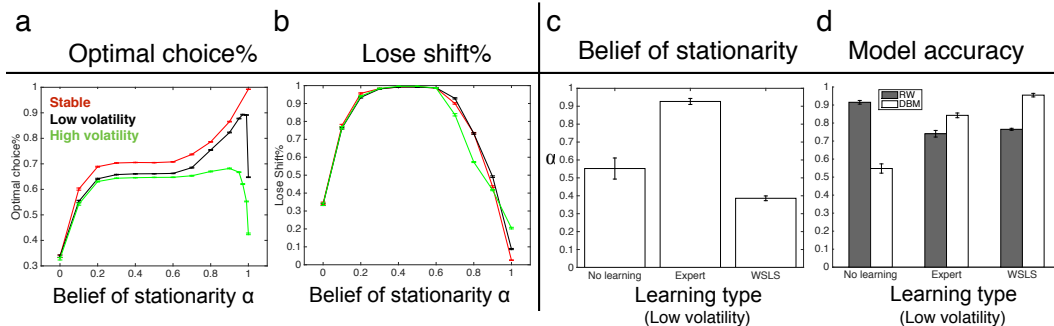

Figure 3: DBM simulation. a. Percentage of trials in which agents chose the optimal choice (the most likely location) as a function of $\alpha$ in three volatility conditions. b: Lose shift rate as a function of $\alpha$ in three volatility conditions. c. $\alpha$ estimation of three types of behavior in low volatility condition ($N(30, 1)$). d. Model performance comparison between R-W and DBM in low volatility condition.

Table 2: DBM: Influence of stationarity belief $\alpha$

| Condition | Decision optimality | Lose-shift rate |
|---|---|---|
| Stable | Positive linear relationship, $\alpha_{optimal} = 1$ | Inverted U shape |
| Low volatility | Inverted U shape, $\alpha_{optimal}$ = high | Inverted U shape |
| High volatility | Inverted U shape, $\alpha_{optimal}$ = medium | Inverted U shape |

## 4 Experiment

We applied the above models to two sets of data in a visual search task: (1) stable condition with no change points (from Yu & Huang, 2014) and (2) low volatility condition with change points based on $N(30, 1)$. For both data sets, we fitted an R-W model and DBM for each subject, and compared learning rate $\eta$ in R-W and estimation of stochasticity $(1 - \alpha)$ in DBM, as well as how they correlate with decision optimality and lose shift rate. For (2), we also looked at how model parameters differ in explaining *No Learning*, *Expert* and *WSLS* behavior.

### 4.1 Results

#### 4.1.1 Stable condition

In a visual search task with relative reward frequency 1:3:9 but no change points, we found a significant correlation between $\eta$ estimated from R-W model and $1$-$\alpha$ from DBM ($r^2 = .84, p = .0001$, Fig 4a), which is consistent with the hypothesis that both the learning rate in R-W model and the belief of stochasticity in the Bayesian approach reflects subjects' estimation of environmental volatility. We also examined the relationship between decision optimality (optimal choice%) (Fig 4b) and lose-shift rate (Fig 4c) with $\eta$ and $1$-$\alpha$ respectively. As is shown in Fig 4b, in this stable condition, decision optimality decreases as the learning rate increases, as well as the belief of stochasticity increases, which is consistent with Fig 2a (red, for $\eta \geq .1$) and Fig 3a (red). For lose-shift rate, there is a significant positive relationship between lose-shift% and $\eta$, as shown previously in Fig 2b (red), and an inverted U shape as suggested in Fig 3b (red). There are no significant differences in the prediction accuracy of R-W model and DBM (R-W: .81+/-.03, DBM: .81+/-.03) or inverse decision parameters ($p > .05$).

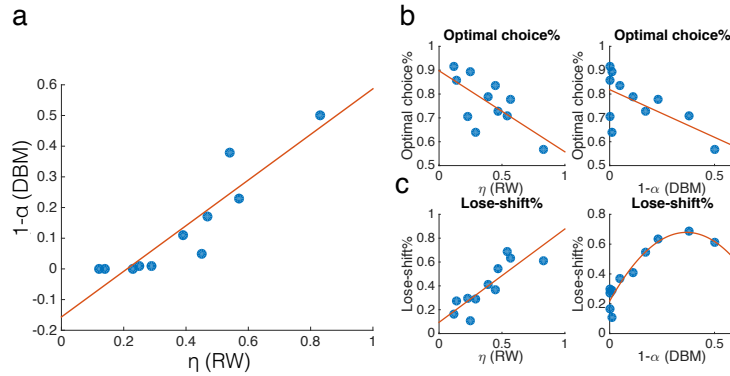

Figure 4: Stable condition: R-W vs. DBM. a. Relationship between learning rate $\eta$ in R-W model and $1$-$\alpha$ in DBM. b. Optimal choice% as a function of $\eta$ and $1$-$\alpha$. c. Lose-shift% as a function of $\eta$ and $1$-$\alpha$.

#### 4.1.2 Low volatility condition

In a visual search task with relative reward frequency 1:3:9, and change of the reward contingency based on $N(30, 1)$ (3 blocks of 90 trials/block), we looked at the correlation between model parameters, their correlation with decision optimality and lose-shift rate, as well as how model parameters differ in explaining different types of behavior.

Subjects (N=207) were grouped into poor learners (optimal choice%$< .5$, n = 63), good learners (.5$\leq$ optimal choice% $\leq 9/13$, n = 108) and expert learners (optimal choice% $> 9/13$, n = 36) based on their performance (percentage of trials started from the most likely rewarded location). Consistent with what we have shown previously (Fig 3d-No Learning), R-W model outperformed DBM in poor learners ($p = .000$). Similar as in stable condition (Fig 4a), among good and expert learners, there is a significant positive correlation between $\eta$ and 1-$\alpha$ (Fig 5b, $r^2 = .35, p = .000$). The relationship between decision optimality and lose-shift% is shown in Fig 5c. As is shown, in this task where change points occur with relatively low frequency ($N(30, 1)$), lose shift% has an inverted U shape as a function of optimal choice%, indicating that a high lose-shift rate does not necessarily lead to better performance.

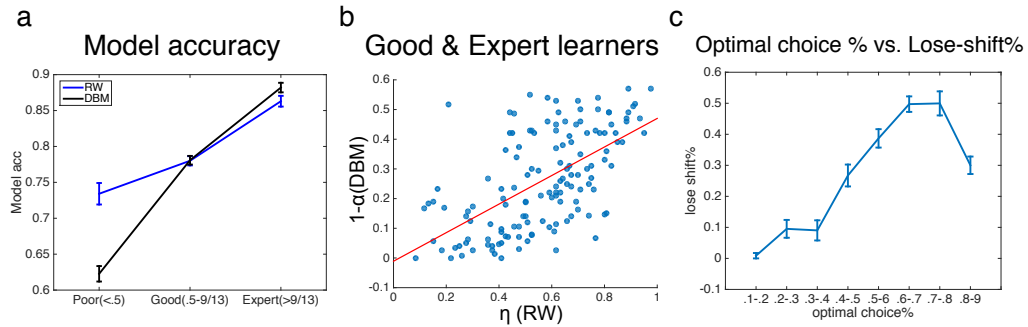

Figure 5: a. Prediction accuracy in poor, good and expert learners. b. Correlation between $\eta$ from R-W and 1-$\alpha$ from DBM. c. Correlation between optimal choice% and lose-shift%.

Next, we looked at how each model parameter correlates with decision optimality and lose-shift rate. For decision optimality (Fig 6ab), consistent with simulation result, it has an inverted U shape as a function of learning rate $\eta$ in R-W model (Fig 6a), while it is positively correlated with $\alpha$ in DBM (Fig 6b). For lose-shift rate (Fig 6cd), also consistent with simulation result, it is positively correlated with $\eta$ in R-W (Fig 6c), while having an inverted U shape as a function of $\alpha$ in DBM (Fig 6d).

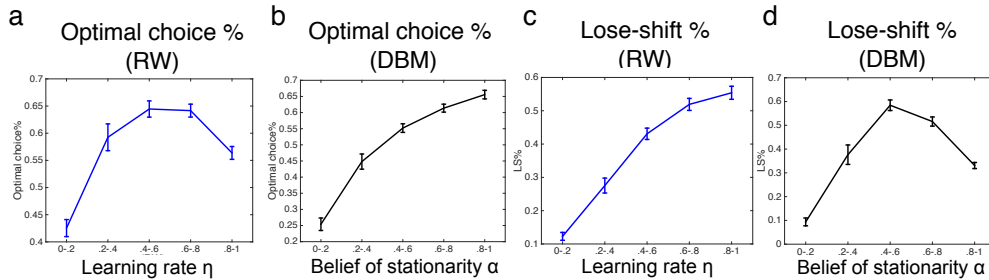

Figure 6: Decision optimality and lose-shift rate. a. Optimal choice% as a function of $\eta$ in R-W model. b. Optimal choice% as a function of $\alpha$ in DBM. c. Lose-shift% as a function of $\eta$ in R-W model. d. Lose-shift% as a function of $\alpha$ in DBM.

In addition, we examined how poor, expert learners and individuals with a high lose-shift rate (*LS*, lose shift%$> .5$ and optimal choice% $< 9/13$, n = 51) differ in model parameters (Figure 7). Consistent with what we have shown (Fig 2c), those three different behavioral types had significantly different learning rate (one-way ANOVA, $p = .000$) and each condition is significant from each other ($p = .000$ for t test across conditions), in which poor learners had the lowest learning rate while subjects with high lose-shift rate had the highest learning rate (Fig 7a). Belief of stationarity from DBM also confirmed what we have shown (Fig 3c), that expert subjects had significantly higher belief of stationarity (one-way ANOVA, p = .003, and $p = .004$ for t test comparing to Poor subjects and $p = .000$ comparing to LS subjects). It also suggested that poor learners did not differ from LS subjects ($p > .05$), though DBM had a lower accuracy in predicting poor learners' choices (Fig 5a). No significant difference of inverse decision parameter $\beta$ was found between R-W and DBM for

expert and LS subjects ($p > .05$), but it was significantly lower in poor learners estimated in DBM (Supplementary Figure S3).

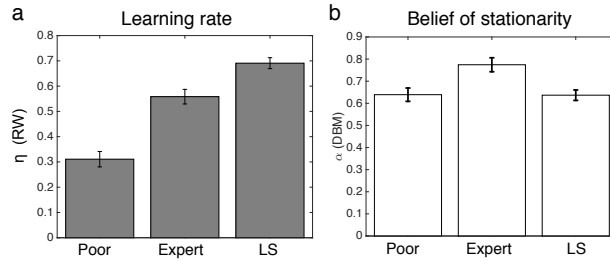

Figure 7: Parameter estimation for different behavioral types. a. Learning rate in R-W model. b. Belief of stationarity in DBM.

## 5 Discussion

In this paper we compared an R-W model and a Bayesian model in a visual search task across different volatility conditions, and examined parameter differences for different types of learning behavior. We have shown in simulation that both the learning rate $\eta$ estimated from R-W and the belief of stochasticity $1 - \alpha$ estimated from DBM have strong positive correlation with increasing volatility, and confirmed that they are highly correlated with behavioral data (Fig 4a and Fig5b). This suggests that both models are able to reflect an individual's estimation of environmental volatility. We also have shown in simulation that R-W model can differentiate No Learning, Expert and WSLS behavioral types with (increasing) learning rate, and DBM can differentiate Expert and WSLS behavioral types with (increasing) belief of stochasticity, and confirmed this with behavioral data in a low volatility condition. A few other things to note here:

**Correlation between decision optimality and lose-shift rate**. Here we have provided a model-based explanation of using lose-shift strategy and how it is related to decision optimality. 1) R-W model suggests that, across different levels of environmental volatility, the frequency of using lose-shift is positively correlated with learning rates (Fig 2b). However, decision optimality is NOT positively correlated with lose-shift rate across conditions. 2) DBM model suggests that, across different levels of environmental volatility, there is an inverted U shape relationship between the frequency of using lose-shift and one's belief of stationarity (Fig 3b), and a close-to-linear relationship between decision optimality and the belief of stationarity in a low volatility environment (Fig 6b).

**Implications for model selection**. We have shown that both models have comparable prediction accuracy for individuals with good performance, but R-W model is better in explaining poor learners' choice. There are several possible reasons: 1) the Bayesian model assumed subjects would use the feedback information to update the posterior probability of target reward distribution. Thus for 'poor' learners who did not use the feedback information, this assumption is no longer appropriate. 2) the R-W model assumed subjects would only update the chosen option's value, thus error trials may have less influence (especially in the early stages, with low learning rate). That is, for 0 value option, it will remain 0 if not rewarded, and for the highest value option, it will remain being the highest value option even if not rewarded. Therefore, R-W model may capture poor learners' search pattern better with a low learning rate.

**Future directions**. For future work, we will modify current R-W model with a dynamic learning rate that will change based on value estimation, and modify current DBM model with a parameter that controls how much feedback information is used in updating posterior belief and a hyper-parameter that models the dynamic of $\alpha$.

### Acknowledgements

We thank Angela Yu for sharing the data in Yu et al. 2014, and for allowing us to use it in this paper.

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
