[Reviews · NeurIPS 2016]

Reviewer 1

Summary

In this paper, the authors considered the learning under uncertainty and compared the R-W model with a fixed learning rate and a Bayesian approach. The author briefly introduced the R-W model under stable, low volatility and high volatility conditions. Then the author compared it with a dynamic brief model that is based on a Bayesian approach. Some simulation is provided for all three conditions that compares these two approaches. A real data example is further considered for illustration.

Qualitative Assessment

I have the following major comments on this paper based on my readings. 1. Simulation seems insufficient, as the plot is zigzag-shaped in Figures 2 and 3. More repetition is preferred. 2. Is the R-W model or Bayesian approach valid in practice? How can we use them to solve real problems? I think more explanation on the real data example in Section 4 would be helpful here, as it is currently not very clear. 3. In real data example, it was mentioned "fitted an R-W model and DBM." However, the authors didn't provide any details about how to fit and implement. I think some discussion is necessary here. If the paper is too long, this part can be moved to Supplement.

Confidence in this Review

2-Confident (read it all; understood it all reasonably well)


Reviewer 2

Summary

The paper compares Rescorla-Wagner and Bayesian models of learning and decision making in environments with different volatility levels. It points out both key similarities and differences between the models and demonstrate how these models can be used to interpret differences between behavioral groups (such as experts, poor learners).

Qualitative Assessment

This is an interesting modeling and model comparison paper, providing insights into the processing of uncertainty during learning and decision making. The paper combines advances that could be interesting to both experimental and modeling audiences. However, its clarity should be improved and parameter estimation details explained much better for the paper to be acceptable to NIPS. More specifically: - Why should highly volatile environments have high learning rates (line 2 of page 2)? Couldn't it plausibly lead to excessive weight instability? - Typo on page 2: "a losS in the current trial". - The R-W model has two parameters (the learning rate and the inverse temperature) but only the former is widely discussed. Are both parameters flexible and estimated in a balanced way (e.g. beta ranging from close to 0 to over 10 with sufficiently many steps)? - What is the exact definition of "expert learner"? Only vague explanation is used. - I would suggest to rename Chapter 3 as "A Bayesian approach". - Why is only the low volatility condition used in Figs. 2c and 3cd? - What exactly is the model accuracy measure? - What is meant by "inverse decision parameters" at the end of 4.1.1? beta? - Why is the 9/13 threshold (and .5 for that matter) chosen to distinguish expert (and poor) learners? Was there any unsupervised learning? As many results are based on these classes, the numbers should be justified or sensitivity to them discussed. - In the fourth last line of page 6 there should be Fig 5c instead of 6c, and fractions, not percentages in the figure. - What is eta_0 in Fig. 6 legend? - Regarding the discussion, as I understand [.5, 0, 0] and [1, 0, 0] would not lead to the same choices if probabilistic action selection (i.e. not very large betas) is used. - as an alternative to both standard RL (with updating only the chosen option's value) and dynamic belief models, RL with eligibility traces could be discussed. - What is the Yu et al., 2014 paper in the Acknowledgments? --- Response to author rebuttals: Thanks for addressing many of my concerns. However, I would like to point out a few things, which I hope can be clarified in the final version. * for model fitting the exact objective function and evaluation of goodness-of-fit should be specified, e.g. maximum likelihood estimation, chi-squared tests * even if estimation results for betas do not show significant differences between groups, it would be useful to have them as well as precise estimation details (such as parameter ranges, steps, methods) at least in the supplementary material for a comparison with other model-based analysis papers, some of which attribute most of behavioral variation to learning rates whereas others to exploration-exploitation.

Confidence in this Review

2-Confident (read it all; understood it all reasonably well)


Reviewer 3

Summary

The paper studies the impact of the learning rate in a reinforcement learning model and belief of stationarity in a Bayesian model on choice in an unstable environment through computer simulations and modelization of experimental data. A special emphasis is put on modelling the behavioral variability observed in experimental studies between non-learners (who basically behave randomly), experts (who always pick up the most rewarded response), and participants between these two extremes who rely on suboptimal win-stay/loose-shift strategy.

Qualitative Assessment

The paper is interesting and well-written though I have a few issues with it. First, the goal of the research was never very clear for me. Is this just an exploration of the effect of learning rate in a RL model and belief of stationarity in a Bayesian model affect performance in an unstable environment? Or is it about seeing whether these models are capturing individual differences in decision-making task in order to see which framework is best suited at a model of human decision making? The paper does not seem to choose between those two goals. I think it would greatly improve the manuscript if the goals were better stated and the results discussed with regard to them. I would recommend the authors to focus on the second of those goals as their data clearly provides a solution. If, for the most part, the learning rate and the belief of stationarity seems to covary, the Bayesian model is actually unable to discriminate between the non-learners and win-stay/loose-shift participants as opposed to the RL model. On this criterion, the RL model provides a better model than the Bayesian approach. Also, even though as I said, the paper is well-written, I believe more details about the simulations are necessary. How many runs? On how many trials are the variable computed? etc. Without those details, it is otherwise sometimes difficult to make sense of the statistics: p-values are most of the time meaningless but they are especially meaningless if there is no idea of the sample size used to compute them. The same way, error-bars are provided for the figures but as there is no mention as to what they are (standard error? confidence interval?), they are of little use. How the accuracy of the models is computed is also not clear. Finally, not much is said about the inverse decision temperature parameter used in the softmax decision rule of the RL model: What was its value during the simulations and when fitting the model to the data? Are the conclusions the same if its value is changed? I have also mixed feelings about Figure 5. Why are the good and expert learners put together? Where are the bad learners? What is the point of the optimal choice vs. lose-shift figure? Finally, some totally superfluous nitpicking: (a) p-values cannot be below 0; (b) the figures sometimes use the belief of stationarity, sometimes its inverse. Why not stick to one of them for the whole paper?; (c) The RL model is more Bush-Mosterler than Rescorla-Wagner (the former applies to Pavlovian conditioning, where no choice is involved and the agent just tries to predict the amount of reward on a given trial, while the other deals indeed with choice betweens several responses).

Confidence in this Review

2-Confident (read it all; understood it all reasonably well)


Reviewer 4

Summary

This paper compares an R-W model (with various constant learning rates) and a Bayesian model (with various constant beliefs of environmental stationarity). Section 2 verifies assumptions about the relation between the learning rate parameter and the performance and behavior of in-silico simulation runs of the R-W model, in three volatility conditions (section 2.1), and for three behavior types (section 2.2). Section 3 verifies assumptions about the relation between the belief of stationarity parameter and the performance and behavior of in-silico simulation runs of the Bayesian model, in three volatility conditions (section 3.1), and for three behavior types (section 3.2). Section 4 compares both models (R-W and Bayesian) "fitted" to simulate experiments performed by (Yu & Huang, 2014) with human subjects in a visual search task.

Qualitative Assessment

You cite (Gershman 2015), but you do not provide an inductive analysis of your experimental results in such an unifying framework. Most of your analysis is deductive or consists in verifying that your simulation results match intuitive reasoning such as "learning rate is expected to increase as the volatility increases" (at line 137). My main comments on the form are related to clarifications of your approach and methods: - A presentation of your simulation method could improve readability. Notably I guess that you have manipulated the R-W learning rate η (i.e. you ran the simulation with various learning rates) and the belief of stationarity α, in which case η and α are simulation inputs; this is not obvious at first reading of "Learning rate is expected to increase as the volatility increases", which suggests that the learning rate is an output of the simulation. - In section §4 "Experiment", whether η and α are inputs or outputs is less clear. Maybe this is due to the "fitting" process mentioned at line 268: in my understanding, fitting consists in running several simulations, setting (inputting) different parameter (η or α) values for each run, then selecting the parameter value which produces the best match with experimental data; hence a given parameter is both an input of each run and an output of the whole fitting process. Maybe a qualitative and a quantitative descriptions of such a fitting process (and how you select the best match) would clarify both your approach and your method. - You should define how you measure "model accuracy" (figure 3 and line 241), or maybe refer to the "Data Analysis: Model Predictive Accuracy" section of (Yu et al. 2014). - At line 268 you mention "subjects", I understand from (Yu et al. 2014) that they are human subjects, but you could recall this fact in your paper. Below are detailed comments on the form: - The text is not aligned with line numbers. In my comments I refer to the line number just above the commented text line. - Line 40: you cite "(Mathys et al. 2011)", but the referenced paper is "(Mathys et al. 2014)". - The last paragraph of section 1 (lines 91-98) refers to two questions. Apparently, the first question starts at line 66, and the second question starts at line 86 ("In addition ..."). Maybe you could highlight the two questions in the next to last paragraph of section 1. - Line 132 "..., with a large β approximates ’maximization’ strategy.": "..., where a large β..."? - Line 160 (or 161) "... agents strictly follow WSLS have ...": "... agents which strictly follow WSLS have ..."? - Line 161 "... indicating their choices are heavily relied on recent experience": is "relied" the correct term here? Do you mean "linked"? Or "... recent experience heavily impacts their choices"? - Line 230-231 "with α = 0 leads to": same comment as line 132. - Line 238-239 "This is consistent with their estimation of higher volatility reflected by higher learning rate than Expert estimated from R-W model (Fig 2c).": I do not understand this sentence. - Line 268 "with change points occur": same comment as line 132. - Line 287 "inverse decision parameters": you define the R-W parameter β at line 132, but you do not define the "inverse decision parameter" used with your DBM simulations. Maybe you should refer to the corresponding parameter β of (Yu et al. 2014 p283-284). - Line 311 "In a visual search task with relative reward frequency 1:3:9 and change of the reward contingency occur based on N(30, 1)": quoting the settings, notably "change of the reward contingency occur based on N(30, 1)", may help readability. - Line 321: "Fig. 6c" should be replaced by "Fig. 5c". - Fig. 5c and lines 321-323: what is/are the subject categorie/s related to fig. 5c? What is/are the model results or human data related to fig. 5c? - Line 356 / Fig. 6a caption: what is "η0"? - Lines 424-425: You should not include acknowledgments in the submitted paper. - Line 370 "with a fixed learning rate and a Bayesian model with a fixed belief of stationarity": these formulation suggest a constant value for each parameter (see my comment above about fitting). You could formulate this way: "with various constant learning rates and a Bayesian model with various constant beliefs of stationarity" Response to author rebuttals: Thanks for addressing many of my concerns. Considering your response to methods clarification requests, I have increased the "clarity" note. Considering other reviewers' qualitative assessment, I've increased the "technical" note. I am less confident about an improvement of novelty and impact criteria. As stated your response, your aim is rather experimental than theoretical.

Confidence in this Review

1-Less confident (might not have understood significant parts)


Reviewer 5

Summary

The article provides a systematic comparison of a model-free (Rescorla-Wagner based) and a model based approach to a visual search task, under varying levels of reward uncertainty. Outcomes of simulated behaviour and real subject choices are provided and discussed. The paper groups real subjects into various categories of learners and discusses their model parameters and implied choice mechanisms.

Qualitative Assessment

The results are generally well presented. However, the presentation is missing a clear presentation of the beta parameter (how many subjects acted how close to random? How meaningfull is the optimal choice for low beta players?) Generally, the nature of the estimation of model parameters for the RW model remains unclear. Was a minimum negative loglikelihood used? While model selection is present in the discussion section, actual model selection criteria are not shown. Post Author Feedback: I thank the authors for clarifying or promising to clarify the above points.

Confidence in this Review

2-Confident (read it all; understood it all reasonably well)


Reviewer 6

Summary

The paper compared the relation between the critical parameters of Roscorla-Wagner model (learning rate) and a Bayesian dynamic belief model (\alpha which quantifies observer's belief of the stationality of the environment), and their relation with the summary statistics (probability of choosing the optimal target and lose-shift rate) of subjects in environments with different volatility, and simulated 3 types of extreme hypothetical subjects and evaluated the properties of the parameters of the two models if they were fitted to such hypothetical subjects.

Qualitative Assessment

The results seem to be solid and the data from real subjects appear compelling. As far as I can tell, the systematic comparison the authors made has not appeared in literature. The conclusion from this paper might contribute to our understanding of what parameters we might expect from an observer in environment of certain volatility, depending on his/her performance. The simulation results in Fig 2a,b and 3a,b provide good match to those observed in Fig 6. However, I feel the theoretical motivation is relatively weak. Providing extra evidence that learning rate can reflect the belief of volatility might be useful to the literature. However, this is not new idea (as in citation 2 and 9). And the \alpha in DBM by definition quantifies the belief of stationality. The summary in Table 1 and 2 may be noteworthy but falls somewhat on the border of novelty. The simulation in section 3.2 appears strange in logic. The No Learning behavior can be caused by many possibility, and one of the most probable cause is a subject who does not care about experiment at all. The Expert behavior who always chooses the optimal target is unrealistic in practice, since no subject could know where the correct target should appear before seeing feedback. Therefore, in my view, 2 out of 3 hypothetical behavioral types in Figure 1 do not appear interesting at all. WSLS might be a valid description of behavior of some subjects but seems better as a theoretical choice strategy which should be considered in parallel with R-W and DBM as models of behavior. I find it hard to consider these as "example of three behavioral types" (Fig 1). I would call at least two of them as hypothetical extreme behaviors. How the model accuracy was calculated was not made clear. The authors should not start using abbreviation for critical terms such as Rescorla-Wagner (R-W) without any introduction to the full term itself. The link of RL with Reinforcement Learning should also be pointed out explicitly for ease of reading for general readers. (new comment after seeing feedback): After taking other reviewers' opinion, I think this paper could have some value to the community. So I upgraded my rating for its potential impact. The author's explanation of expert learner makes a bit more sense to me. But it still sounds like half a god. It basically says the expert know exactly when reward probability changes. To me, a better definition of an "expert" would be an observer who has learnt the exact distribution of hidden parameters of a generative model and makes inference in a statistical optimal way. In the simulated case, the change point occurs randomly following a distribution. There is no way for any observer to learn exact the time when a change point occurs if the the change point itself does not occur at deterministic time points.

Confidence in this Review

2-Confident (read it all; understood it all reasonably well)